# Considering Both GLS and MD for a Prognostic Value in Non-ST-Segment Elevated Acute Coronary Artery Syndrome

**DOI:** 10.3390/diagnostics13040745

**Published:** 2023-02-16

**Authors:** Ioana Ionac, Mihai-Andrei Lazăr, Raluca Șoșdean, Cristina Văcărescu, Marius Simonescu, Constantin-Tudor Luca, Cristian Mornoș

**Affiliations:** 1Cardiology Department, “Victor Babes” University of Medicine and Pharmacy, 300041 Timisoara, Romania; 2Cardiology Department, Institute of Cardiovascular Diseases, 300310 Timisoara, Romania

**Keywords:** 2D strain echocardiography, global longitudinal strain, mechanical dispersion, prognostic, non-ST-segment elevated acute coronary syndrome, cardiac events

## Abstract

Global longitudinal strain (GLS) and mechanical dispersion (MD), as determined by 2D speckle tracking echocardiography, have been demonstrated to be reliable indicators of prognosis in a variety of cardiovascular illnesses. There are not many papers that discuss the prognostic significance of GLS and MD in a population with non-ST-segment elevated acute coronary syndrome (NSTE-ACS). Our study objective was to examine the predictive utility of the novel GLS/MD two-dimensional strain index in NSTE-ACS patients. Before discharge and four to six weeks later, echocardiography was performed on 310 consecutive hospitalized patients with NSTE-ACS and effective percutaneous coronary intervention (PCI). Cardiac mortality, malignant ventricular arrhythmia, or readmission owing to heart failure or reinfarction were the major end points. A total of 109 patients (35.16%) experienced cardiac incidents during the follow-up period (34.7 ± 8 months). The GLS/MD index at discharge was determined to be the greatest independent predictor of composite result by receiver operating characteristic analysis. The ideal cut-off value was −0.229. GLS/MD was determined to be the top independent predictor of cardiac events by multivariate Cox regression analysis. Patients with an initial GLS/MD > −0.229 that deteriorated after four to six weeks had the worst prognosis for a composite outcome, readmission, and cardiac death according to a Kaplan–Meier analysis (all *p* < 0.001). In conclusion, the GLS/MD ratio is a strong indicator of clinical fate in NSTE-ACS patients, especially if it is accompanied by deterioration.

## 1. Introduction

Acute coronary syndrome (ACS) is responsible for a significant part of all deaths caused by cardiovascular disease and is an important contributor to morbidity and mortality during and after hospitalization [1]. Non-ST-segment elevated acute coronary syndrome (NSTE-ACS) shows a poorer prognosis and a higher incidence than ST-segment elevated acute coronary syndrome (STEMI) [1,2]. Risk stratification in these patients is important to correctly undergo an intensified treatment strategy and to prevent unnecessary readmissions to the hospital [1,3]. Assessing left ventricular (LV) remodeling, which leads to heart failure (HF), can predict the prognosis of these patients [4]. Before presenting HF symptoms, patients can develop asymptomatic diastolic and/or systolic LV disfunction, with structural or functional cardiac abnormalities as a cause [5]. Therefore, cardiac imaging is indispensable for managing and following these patients. Prognosis in cardiovascular disease is strongly related to systolic function, commonly assessed using left ventricular ejection fraction (LVEF), and to diastolic function, evaluated by the estimation of left ventricular filling pressures (LVFPs).

When evaluating LVFPs in individuals with cardiac illness, there are numerous criteria to consider. Tissue Doppler imaging (TDI) is a well-established and powerful predictor of unfavorable outcome in several cardiovascular disorders. According to studies [3,6,7,8,9,10], the ratio of early transmitral flow velocity to early diastolic mitral annulus velocity (E/e’) has a significant predictive value for worse outcomes following acute myocardial infarction. Additionally, because longitudinally orientated cardiac fibers are known to be the most sensitive to ischemia, the systolic mitral annulus velocity (s’) is connected to these fibers and can be utilized to evaluate regional motion impairment [5,6,10].

It has been shown that 2D speckle tracking echocardiography (2D-STE) with measurement of the global longitudinal strain (GLS) and of the mechanical dispersion (MD) of the LV can be useful for predicting adverse outcomes in ACS [6,11]. An important advantage of 2D-STE measurements over the Doppler-based technique is its independence of the ultrasound insonation angle [12,13,14]. It has been shown that GLS is a powerful independent predictor of LV remodeling after revascularization therapy in patients with NSTE-ACS [6]. Studies of patients with ischemic and nonischemic cardiomyopathy have shown that a decrease in GLS is linked to an increased risk of ventricular arrhythmias and major cardiac adverse cardiovascular events [15].

MD is also a marker derived from 2D-STE and reflects contraction heterogeneity. MD is described as a predictor of ventricular arrhythmias independently of LVEF and is significantly associated with sudden cardiac death [16,17]. Physiologically, all myocardial segments have, to some extent, similar contraction duration; thus, the values for MD are normally low. Its measurement is a parameter that can be used to predict mortality and ventricular arrhythmias in myocardial infarction, nonischemic cardiomyopathy, HF, and cardiac resynchronization therapy [9,10,16,18,19]. A greater MD was detected in patients with ventricular arrhythmias with ischemic and nonischemic cardiomyopathy than in those that did not present ventricular arrhythmias [20]. Prolonged MD was associated with worse long-term outcomes in patients with STEMI [18].

LV function has been evaluated with 2D-STE measurements in patients with ACS; however, very few data are available in the literature. There are even fewer studies that have investigated GLS and MD measured by 2D-STE in an NSTE-ACS population.

Our group proposed a new index, GLS/MD, to predict cardiac events in patients with previous NSTE-ACS. We aim to examine the relationship between the GLS/MD ratio and cardiac events and the value of GLS/MD worsening during follow-up in an NSTE-ACS patient population after undergoing successful percutaneous coronary intervention (PCI) therapy.

## 2. Materials and Methods

### 2.1. Study Population

The Timisoara Institute of Cardiovascular Diseases’ Cardiology Department serves as an invasive center for 15 non-invasive cardiology units. A total of 2773 patients were admitted to the department for PCI from January 2018 to May 2019. All of these patients were registered in a clinical database. In accordance with the 2015 ESC Guidelines for the management of acute coronary syndromes in patients presenting without persistent ST-segment elevation [21], we prospectively examined 402 consecutive NSTE-ACS patients who underwent successful PCI in sinus rhythm while being hospitalized in our clinic. Residual stenosis of less than 20% was considered a successful PCI; non-culprit intervention was carried out concurrently with hospitalization. Patients with inadequate echocardiographic images, prior myocardial infarction, open-chest surgery, cardiac pacemaker/defibrillator, mitral stenosis, severe primary MR, significant annular calcification, renal failure (serum creatinine > 1.3 mg/dL), and non-cardiac illnesses with a life expectancy of less than a year were excluded from the study group. The study population consisted of the remaining 310 patients. The Institute of Cardiovascular Diseases Timisoara’s Institutional Ethics Committee accepted the study, which was carried out in accordance with the Declaration of Helsinki’s principles. All participants gave their informed consent.

### 2.2. Recorded Clinical Variables

Age, sex, mean arterial pressure, heart rate, body mass index, peak high-sensitivity cardiac troponin I level (hs-cTnI), and N-terminal pro-brain natriuretic peptide (NT-proBNP) levels were clinical characteristics that were noted and incorporated into the predictive model. The main treatment classes that were prescribed were also noted.

For the purposes of this study, the following five cardiovascular risk factors were taken into account: hypertension (systolic blood pressure > 140 mmHg, diastolic blood pressure > 90 mmHg, or in drug treatment), cardiovascular disease heredity, smoking (more than one cigarette per day; cessation of smoking less than 10 years ago was still considered smoking), diabetes mellitus (fasting glycemia > 126 mg/dL or in drug treatment), and hypercholesterolemia (>200 mg/dL or in drug treatment).

### 2.3. Echocardiography

After PCI, an echocardiogram was performed utilizing a Vivid 9 system (General Electric, Milwaukee, WI) at the time of hospital discharge. According to the most recent recommendations, the left atrial volume (LAV) and the indexing LAV to the body surface area (LAVI) were calculated [22]. A modified version of Simpson’s formula was used to compute LVEF from apical two- and four-chamber images [22]. The mitral regurgitation’s regurgitant orifice area (ROA) and regurgitant volume (RV) were calculated [23]. An axial size 3–5 mm pulsed-sample Doppler volume was positioned between the mitral valve tips, and a four-chamber view was used to record the transmitral flow. Peak (E) and late transmitral flow (A) were assessed during end-expiratory apnoea for five consecutive cardiac cycles, and the findings were averaged [24]. The peak velocity of tricuspid regurgitation was used to determine systolic pulmonary artery pressure (SPAP). The pulsed-wave Doppler mode of the TDI program was selected. A 4–5 mm sample volume was successively placed at the septal and lateral corners of the mitral annulus in the apical four-chamber view [24]. Peak e’ and s’ were measured during end-expiratory apnoea for five consecutive cardiac cycles, and the data were averaged [7]. The average velocities of the septal and lateral sites were used to calculate E/e’.

The apical four-chamber, apical two-chamber, and apical three-chamber views were captured in two dimensions. For each view, three cardiac cycles were recorded. Aortic valve closure was determined from conventional pulsed-wave Doppler signals through the aortic valve. Peak systolic strain was measured across all LV segments, and GLS was calculated by averaging the values from each segment. The MD was calculated as the average time between the peak of the R wave and the peak of the negative strain in various LV segments [25]. Segments with strain curves swinging about the zero line and segments with solely positive strain values, known as akinetic segments and dyskinetic segments, respectively, were excluded [18]. Patients were eliminated from the study if more than three LV segments had insufficient tracking or if six or more segments lacked satisfactory monitoring. Next, the GLS/MD ratio was determined.

Four to six weeks after leaving the hospital, measurements were taken again.

For GLS, MD, and GLS/MD, the inter- and intraobserver variability was investigated. In 35 randomly chosen subjects of the study group, two investigators independently recorded measurements a few minutes apart in the absence of the other investigator. Both observes were blinded to each other’s result.

### 2.4. Clinical Outcome

The patients were monitored for about 24 months. Cardiac mortality, malignant ventricular arrhythmias, hospital readmission for HF, and reinfarction were the main events. Deaths specifically linked to a cardiac condition, primarily congestive heart failure, reinfarction, or sudden cardiac death, were considered cardiac deaths. The follow-up data were gathered through computerized medical records or by calling the patients or their relatives.

### 2.5. Statistical Analysis

For continuous variables, data were presented as means plus standard deviation (SD), while for categorical variables, they were presented as proportions. An unpaired *t*-test (for variables with a normal distribution) or Mann–Whitney U test was used to compare continuous variables between groups (non-normally distributed variables). With the help of the χ^2^ test and Fischer’s exact test, proportions were compared. Using a two-sample *t*-test, continuous variables were compared between groups with cardiac events vs. non-cardiac events (variables with normal distribution). Area under the ROC curves (AUC) were compared after receiver operating characteristic (ROC) curves for prediction of cardiac events were established for various parameters. At the time of death, patients who passed away from non-cardiac reasons were censored.

Cox regression models with single and multiple variables were developed to link echocardiographic results to the main outcome. To examine the predictive power of baseline predictors, the data from the univariable Cox regression were incorporated into the multivariable Cox regression. Kaplan–Meier analysis was used to determine the cardiac event-free survival rates, and a log-rank test was used to compare the event rates. A *p*-value of 0.05 or less was regarded as significant.

Intraclass correlation coefficients were used to quantify the intraobserver variability and interobserver variability for GLS, MD, and GLS/MD. Data were analyzed utilizing SPSS Statistics for Windows (IBM SPSS Statistics for Windows, Version 21.0. Armonk, NY, IBM).

## 3. Results

This prospective analysis comprised 310 consecutive patients hospitalized for NSTE-ACS from January 2018 to May 2019 who received successful PCI. The institution’s guidance was followed for the routine treatment of all patients. The average age of our study sample was 60.2 ± 11.7 years, and 70.3% of the participants were men (218 patients). No patient was lost to follow-up in this trial. A total of 109 patients (35.16%) experienced cardiac incidents during the follow-up period (34.7 ± 8 months). Cardiovascular mortality occurred in 6 patients (1.93%), and readmission occurred in 103 patients (33.2%) as the first cardiac event. Among our patients, 28 (9%) experienced non-fatal myocardial ischemia, whereas 75 (24.2%) required hospitalization for heart failure. A total of 29 patients (9.35%) experienced cardiac death over the course of the follow-up.

### 3.1. Patient Characteristics

Table 1 lists the initial characteristics of the patients. Higher NT-proBNP levels, SPAP, E, E/A, E/e’ ratio, GLS, and GLS/MD ratio and greater surface and diameter of the LA, LAV, and LAVI, as well as end-diastolic LV diameter and longer MD, were seen in patients who experienced cardiac events. Their LVEF, A, e’, and s’ velocities were decreased. There was also no difference in the distribution of the following factors: gender, body mass index (BMI), previous coronary artery disease, cardiovascular risk factors (diabetes, smoking, hypertension, heredity, and dyslipidaemia), presence of severe mitral regurgitation (MR), levels of peak high-sensitivity cardiac troponin I, medication (i.e., beta blockers, diuretics, and nitrates), E-deceleration time, mitral regurgitant orifice area, or mitral regurgitant volume. Patients without incidents had a mean GLS/MD at discharge of −0.418 ± 0.2, while those who had an incident had a mean GLS/MD at discharge of −0.239 ± 0.13 (*p* < 0.001).

### 3.2. ROC Curves to Predict Cardiac Events

The most accurate echocardiographic parameters for forecasting cardiac events are shown in Figure 1 as ROC curves. The maximum accuracy for the GLS/MD index was determined by the area under the ROC curve (AUC = 0.849, 95%CI = 0.805–0.893, *p* < 0.001). The baseline E/e’ ratio, MD, and GLS (AUC = 0.794, 95%CI= 0.741–0.847, *p* < 0.001; AUC = 0.738, 95%CI = 0.684–0.792, *p* < 0.001; and AUC = 0.652, 95%CI = 0.587–0.717, *p* < 0.001, respectively) were significant for prediction of composite outcomes. Statistical comparison of the ROC curves shows substantial differences between GLS/MD and MD (*p* = 0.029) and between GLS/MD and E/’e (*p* = 0.015). The AUC was lower for every other echocardiographic parameter that was examined. The composite outcome can be predicted using GLS/MD at discharge with an appropriate cut-off value of −0.229 (82% sensitivity and 73% specificity).

### 3.3. Predictors of Outcome

Table 2 lists the echocardiographic factors (*p* < 0.05) that predicted cardiac events in a univariate Cox regression analysis: LAV, LAVI, LVEF, SPAP, ROA, E, A, E/A, e’, s’, E/e’, GLS, MD, and GLS/MD. On the other hand, a univariate study revealed no significant associations between cardiac events and E-deceleration time, RV, or TAPSE. Then, to track the occurrence of the cardiac events, LAVI, LV end-diastolic volume index, LVEF, SPAP, ROA, E/A, s’, E/e’, GLS, MD, and GLS/MD were all included in a multivariate analysis. The greatest independent echocardiographic predictor of composite outcome was GLS/MD before discharge (HR = 3.621, 95%CI = 2.167–5.075, *p* < 0.001).

### 3.4. Worsening of GLS/MD Ratio during Follow-Up

We found that in 100 patients (30.1%), four to six weeks after hospital release, the GLS/MD ratio had worsened. A total of 33 (33%) of these patients had a GLS/MD starting value greater than −0.229. However, as seen in Figure 2, regardless of the GLS/MD value at study inclusion, GLS/MD worsening was associated with lower event-free survival rates (18.2% versus 35.1% in patients with initial GLS/MD > −0.229 and 76.1% vs. 88% in those with GLS/MD ≤ −0.229 at hospital discharge, respectively; log-rank, *p* < 0.001). The worst prognosis was shown for the composite outcome (Figure 2a) of cardiac mortality (Figure 2b), and ventricular arrhythmia (Figure 2c), which occurred during follow-up in the subgroup of patients with an initial GLS/MD ratio > −0.229 and deteriorating after 4–6 weeks. Regardless of the severity of the condition, the group of patients with baseline GLS/MD > −0.229 showed a greater likelihood of hospital readmission (Figure 2d).

### 3.5. Reproducibility

The intra- and interobserver agreements were good for GLS/MD, GLS, and MD measurements in 35 patients with ischemic heart disease. For interobserver and intraobserver variability, the intraclass correlation coefficients were 0.92 and 0.89 for GLS/MD, 0.94 and 0.93 for GLS, and 0.92 and 0.90 for MD, respectively.

## 4. Discussion

To the best of our knowledge, this is the first study to investigate the value of this index to predict cardiac events (cardiac death, ventricular arrhythmia, and rate of hospital readmission) in patients with NSTE-ACS who were successfully treated by PCI. In our work, GLS/MD provided the best independent echocardiographic prediction of composite outcome, showing the highest accuracy. Patients with an initial GLS/MD ratio >−0.229 and worsening after four to six weeks presented the worst prognosis regarding composite outcome, cardiac death, and ventricular arrhythmias. The group of patients with an initial GLS/MD ratio >−0.229 presented a higher occurrence of hospital readmission, irrespective of its worsening. We investigated the prognostic value of a two-dimensional strain index combining a parameter that evaluates the systolic function and is a measure of infarct size, i.e., GLS, and a parameter that can predict ventricular arrhythmias in patients with different cardiac diseases and is a measure of myocardial deformation heterogeneity, i.e., MD [26].

Predicting prognosis in patients with NSTE-ACS is of very high importance. Risk stratification and identification of high-risk NSTE-ACS patients improves the effectiveness of care because these patients benefit from an intensified treatment strategy [3,8,27]. Echocardiography plays a central role in assessing LV function after an ACS and in identifying HF diagnosis and development, as well as for follow-up [20,23].

Our investigation revealed no differences in the distribution of gender, prior cardiovascular disease, or cardiovascular risk factors between the groups with and without cardiac events, in contrast to what is reported in the literature [5]. Myocardial injury and LV filling pressure, which have an effect on LA, are connected to the prognosis of patients with NSTE-ACS. The elements that determine diastolic LV filling have a significant impact on LA size. LA size, LV volume indices, and LVEF have been shown to be reliable predictors for the monitoring of cardiovascular risk and direction of therapy in patients with ACS in prior research using standard echocardiographic imaging [6,20,28,29]. One of the best indicators of survival in ACS patients is an echocardiographic assessment of LV function before hospital discharge [30].

In our study, LAVI, SPAP, and E/A ratio were eliminated as predictors of outcome in univariate analysis on multivariate analysis.

TDI has an incremental prognostic value for routine clinical, laboratory, and imaging information; it is widely used, with proven easy accessibility and good reproducibility. Unlike LV volume and LVEF, pulsed TDI does not require tracing of the endocardial contours. The E/e’ ratio is a good Doppler predictor for evaluation of the LV filling pressure and can be used as a prognostic marker of cardiac outcome after acute coronary syndrome [7,29]. Its prognostic value has also been demonstrated in patients with NSTE-ACS who underwent PCI [9]. The superiority of TDI parameters could be explained by the dependence of the mitral flow on LA pressure, volume status, myocardial relaxation, and age. In patients with myocardial ischemia, e’ and s’ velocities have been observed to be significantly decreased, but cut-off values that could detect significant coronary disease are still not clear [6,10,27]. However, TDI has some technological limitations, such as angle dependence, signal noise, and measurement variability.

Current guidelines recommend repeated echocardiographic examination for patients with an ACS and severe LV systolic dysfunction after six to twelve weeks, using only LVEF to assess LV systolic function [20]. Thus, LVEF can be preserved in the presence of HF, and GLS and MD are not yet recommended when assessing systolic function [22]. However, it has been demonstrated that GLS provides superior prognostic information regarding the risk of HF in patients with ACS compared to other echocardiographic measurements and that it can be used as an accurate marker of LV function [5,31,32,33,34]. GLS is a well-validated marker for measuring LV longitudinal deformation and is superior to LVEF [35,36,37,38,39,40,41,42,43]. GLS has been shown to be a more sensitive and earlier marker of cardiac disfunction than conventional echocardiography, detecting subclinical abnormalities of both systolic and diastolic function [5,11,44]. GLS significantly correlates with global infarction size and is superior to LVEF in identifying small and medium-sized infarcts [45].

Moreover, in STEMI patients, MD was shown to have a relationship with systolic and diastolic function (GLS and E/e’), infarct size, and electrical dispersion (QRS duration), all of which are strong independent predictors of outcome in HF [46,47,48,49,50]. In addition, MD was shown to be a strong independent predictor of arrhythmic events, and GLS provides information for the risk stratification [16]. It has been shown that that 2D-STE strain can be superior to LVEF in assessing myocardial function after an ACS [5] and that GLS can be a measurement for risk stratification in patients with relatively preserved EF [51].

We re-evaluated GLS, MD, and the GLS/MD ratio after four to six weeks, and our result showed that a GLS/MD ratio > −0.229 before hospital discharge and its worsening in the next weeks is a strong independent predictor of an increased risk of future cardiac events (cardiac death and ventricular arrhythmias) and hospital readmission.

However, our investigation is subject to a number of limitations. Its small sample size is the most significant. Larger investigations are therefore required to confirm our findings.

## 5. Conclusions

In conclusion, our findings support the fact that 2D-STE could be used in patients with NSTE-ACS who have undergone successful PCI. The GLS/MD ratio can be an important independent prognosis factor of cardiac events (cardiac death, malignant ventricular arrhythmia, or hospital readmission for HF and/or reinfarction). Patients with an initial GLS/MD > −0.229 that worsened after a four- to six-week presented the worst prognosis regarding composite outcome, hospitalization, and ventricular arrhythmias. Moreover, patients with an initial GLS/MD ratio > −0.229 presented a higher occurrence of hospital readmission, irrespective of its worsening. This result may have important implications in stratifying risk in patients with NSTE-ACS.

## Figures and Tables

**Figure 1 diagnostics-13-00745-f001:**
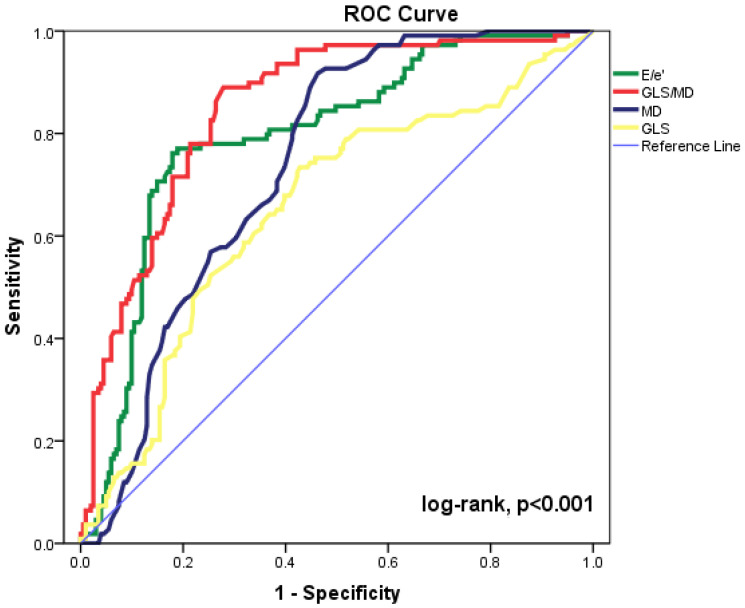
Receiver operating characteristic (ROC) curves for GLS/MD, E/e’ ratio, MD, and GLS to predict cardiac events in our patient group. CI = confidence interval; E = peak early diastolic transmitral velocity; e’ = peak early diastolic mitral annular velocity; GLS = global longitudinal strain; MD = mechanical dispersion.

**Figure 2 diagnostics-13-00745-f002:**
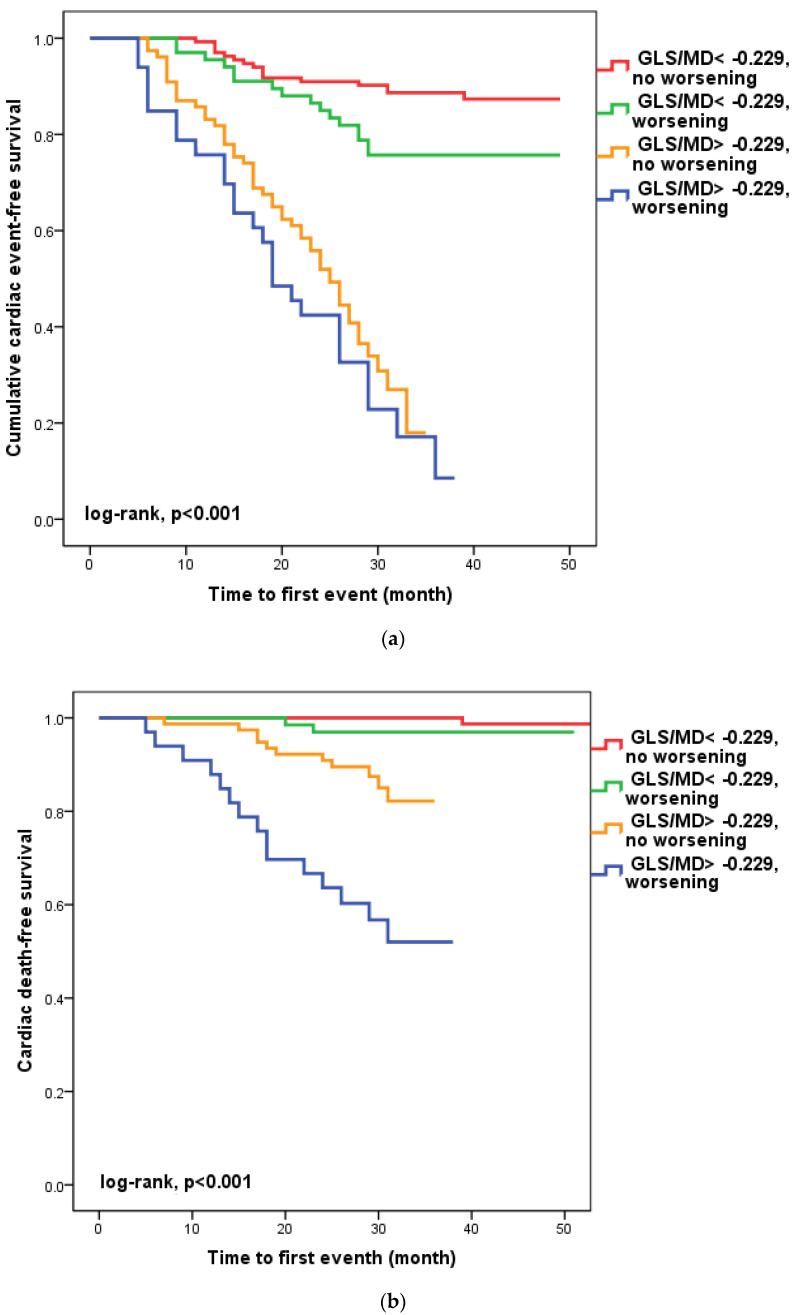
Kaplan-Meier survival curves of composite outcome (**a**), cardiac death (**b**), ventricular arrythmia (**c**), and hospital readmission (**d**) according to the initial GLS/MD value below and above −0.229 and worsening after hospital discharge. GLS = global longitudinal strain; MD = mechanical dispersion.

**Table 1 diagnostics-13-00745-t001:** Baseline characteristics of the study groups.

Characteristic	Event-Free (*n* = 201)	Cardiac Events (*n* = 109)	*p*-Value
Clinical characteristics			
Age, years Female/male gender	60.5 ± 11.7 56/145	59.4 ± 12.1 36/73	0.472 0.36
Body mass index, kg/m^2^	26.8 ± 4.57	27.4 ± 4.66	0.248
Mean arterial pressure, mmHg Heart rate, beats/min Previous coronary artery disease, *n* (%)	97.9 ± 12.8 84 ± 18 47 (23.4)	96 ± 14.5 89 ± 21 31 (28.4)	0.261 0.285 0.34
Hypercholesterolemia, *n* (%)	142 (70.6)	69 (63.3)	0.81
Current smoker, *n* (%)	167 (83.1)	89 (81.6)	0.76
Diabetes mellitus, *n* (%)	99 (49.2)	44 (40.4)	0.15
Systemic hypertension, *n* (%)	77 (38.3)	44 (40.3)	0.41
Family history of cardiovascular disease, *n* (%)	49 (24.4)	25 (22.9)	0.89
Severe mitral regurgitation, *n* (%)	31 (15.4)	25 (22.9)	0.122
NYHA I, *n* (%) NYHA II, *n* (%) NYHA III, *n* (%) NYHA IV, *n* (%)	118 (58.7) 72 (35.8) 6 (2.98) 5 (2.48)	36 (33) 57 (52.3) 14 (12.8) 2 (0.9)	0.004 0.007 0.002 0.011
Laboratory finding			
NT-proBNP, pg/mL	1274 ± 1667	2705 ± 2516	<0.001
Peak high-sensitivity cardiac troponin I, ng/L	519 ± 1256	628 ± 1235	0.464
Culprit lesion			
Left anterior descending, *n* (%)	66 (32.8)	20 (18.3)	0.008
Circumflex artery, *n* (%)	37 (18.4)	20 (18.3)	1
Right coronary artery, *n* (%)	71 (35.3)	58 (53.2)	0.003
Left main stem coronary artery, *n* (%)	27 (13.4)	11 (10.1)	0.47
Multivessel lesion, *n* (%)	48 (23.9)	28 (25.7)	0.78
Therapy at hospital discharge			
Beta blocker, *n* (%) ACEI/angiotensin receptor antagonist, *n* (%)	179 (89.0) 196 (97.5)	101 (92.6) 105 (96.3)	0.42 0.81
Diuretics, *n* (%) Calcium blocker, *n* (%)	153 (76.1) 47 (23.3)	79 (72.4) 26 (24.5)	0.49 0.85
Nitrates, *n* (%) Aspirin, *n* (%) P2Y12 inhibitor, *n* (%) Statin, *n* (%)	141 (70.1) 201 (100) 201 (100) 198 (98.5)	69 (63.3) 109 (100) 109 (100) 105 (96.3)	0.25 1 1 0.98
Echocardiographic indices at hospital discharge			
End-diastolic LV diameter, cm/m^2^	2.9 ± 0.5	3.3 ± 0.6	<0.001
LV ejection fraction, %	49 ± 12	36 ± 12	<0.001
Left atrial volume, mL	79 ± 30	111 ± 50	<0.001
Indexed left atrial volume, mL/m^2^	42 ± 17	58 ± 28	<0.001
Left atrium diameter, cm	4.4 ± 0.5	4.9 ± 0.6	<0.001
Left atrium surface, cm^2^	24 ± 6	29 ± 7	<0.001
Systolic pulmonary artery pressure, mmHg	38 ± 10	44 ± 11	<0.001
Mitral regurgitant orifice area, mm^2^	24 ± 8	26 ± 11	0.046
Mitral regurgitant volume, mL	36 ± 15	39 ± 14	0.253
E, cm/s	73 ± 25	88 ± 29	<0.001
A, cm/s	87 ± 36	72 ± 34	<0.001
E/A ratio	0.94 ± 0.44	1.54 ± 1.03	<0.001
E-deceleration time, ms	173 ± 72	164 ± 74	0.30
e’, cm/s	8.9 ± 3.3	6.5 ± 2	<0.001
E/e’ ratio	8.8 ± 3.5	13.9 ± 4	<0.001
s’, cm/s	8 ± 2.7	5.2 ± 1.8	<0.001
GLS, %	−18.9 ± 5.6	−16 ± 5.5	<0.001
MD, msec	52.85 ± 22	70 ± 16	<0.001
GLS/MD	−0.418 ± 0.2	−0.239 ± 0.13	<0.001

A = late transmitral flow velocity; ACEI = angiotensin-converting enzyme inhibitor; E = early diastolic transmitral flow velocity; e’ = early mitral annular diastolic velocity; GLS = global longitudinal strain; LV = left ventricle; MD = mechanical dispersion; Non-STEMI = non-ST elevation acute myocardial infarction; NT-proBNP = N-terminal pro-brain natriuretic peptide; NYHA = New York Heart Association; s’ = systolic velocity of mitral annulus.

**Table 2 diagnostics-13-00745-t002:** Echocardiographic variables at hospital discharge associated with composite end points (hospital readmission, ventricular arrythmia, or cardiac death) in Cox univariate and multivariate analysis.

Variable	Univariate HR (CI 95%)	*p*-Value	Multivariate HR (CI 95%)	*p*-Value
LV end-diastolic volume index	2.612 (1.878–3.631)	<0.001	1.039 (0.645–1.676)	0.874
LVEF	0.931 (0.914–0.948)	<0.001	0.968 (0.945–0.990)	0.006
Left atrial volume	1.018 (1.013–1.024)	<0.001		
Indexed left atrial volume	1.065 (1.045–1.086)	<0.001	1.005 (0.996–1.014)	0.291
SPAP	1.042 (1.026–1.058)	<0.001	0.988 (0.971–1.006)	0.184
Mitral regurgitant orifice area	1.020 (1.001–1.038)	<0.001	1.019 (0.997–1.041)	0.090
Mitral regurgitant volume	1.008 (0.996–1.020)	0.219		
TAPSE	1.007 (0.996–1.017)	0.228		
E velocity	1.016 (1.010–1.023)	<0.001		
E-deceleration time	0.998 (0.996–1.001)	0.213		
A velocity	0.985 (0.977–0.993)	<0.001		
E/A ratio	1.948 (1.653–2.295)	<0.001	1.017 (0.743–1.393)	0.914
e’ velocity	0.765 (0.703–0.833)	<0.001		
E/e’ ratio	1.180 (1.143–1.219)	<0.001	1.190 (0.954–1.484)	0.124
s’ velocity	0.569 (0.498–0.649)	<0.001	0.763 (0.654–0.891)	0.001
GLS	1.091 (1.054–1.128)	<0.001	1.010 (0.950–1.074)	0.745
MD	1.030 (1.021–1.039)	<0.001	1.009 (0.994–1.025)	0.248
GLS/MD ratio	3.718 (1.239–1.6.197)	<0.001	3.621 (2.167–5.075)	0.001

A = peak late diastolic transmitral velocity; CI = confidence interval; E = peak early diastolic transmitral velocity; e’ = peak mitral annular diastolic velocity; GLS = global longitudinal strain; HR = hazard ratio; LV = left ventricle; LVEF = left ventricular ejection fraction; MD = mechanical dispersion; s’ = peak systolic velocity of mitral annulus; SPAP = systolic pulmonary artery pressure; TAPSE = tricuspid annular plane systolic excursion.

## Data Availability

The datasets used and analyzed during the current study are available from the corresponding author upon reasonable request.

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
