# Peer review of "Considering Both GLS and MD for a Prognostic Value in Non-ST-Segment Elevated Acute Coronary Artery Syndrome"

_diagnostics, 2023, doi:10.3390/diagnostics13040745_

Round 1

Reviewer 1 Report

This is a cohort study that evaluated the role of 2D-speckle tracking echocardiography, namely GLS and mechanical dispersion indices, in predicting the prognosis in NSTE-ACS patients who underwent successful PCI. The study examined the association between a proposed ratio of GLS/MD with a composite of outcomes which included cardiac mortality, malignant ventricular arrhythmias, hospital readmissions for HF, or re-infarction. The results revealed a statistically significant association, which led to the suggestion of a probable prognostic utility of this parameter in the specified patient population. Overall, the study is well written and appears to have a more-or-less sound methodological approach. Nonetheless, few points are worth mentioning here.

2.1. Study Population

“We prospectively examined 402 consecutive patients NSTE-ACS who underwent successful PCI in sinus rhythm while being hospitalized in our clinic.”

What is the rationale behind mandating the presence of an underlying sinus rhythm? 

“Patients with inadequate echocardiographic images, prior myocardial infarction, open-chest surgery, cardiac pacemaker/defibrillator, significant valvular heart disease, renal failure (serum creatinine >1.3 mg/dL), and non-cardiac illnesses with a life expectancy of less than a year were excluded from the study group.”

Please specify what was considered a “significant” valvular heart disease in this study as this has important implications from an echocardiographic evaluation standpoint. Also, if these patients were excluded from the study, how did 31 patients in the event free group and 25 patients in the cardiac events group get into the study even though they had severe mitral regurgitation per table 1?

2.2. Clinical Variables Recorded

“Age, sex, mean arterial pressure, heart rate, body mass index, peak high sensitivity cardiac troponin I level (hs-cTnI), and N-terminal pro-brain natriuretic peptide (NT-proBNP) levels were clinical characteristics that were noted and incorporated into the predictive model.”

Age, mean arterial pressure, and heart rate are not reported in the results text or table, which raises the question about how was this data incorporated in the analysis.

“The main treatment classes that were prescribed were also noted.”

What are the main treatment classes here? Is this referring to beta blockers, diuretics, and nitrates? If so, why were those –and not antiplatelet therapy, statin, ACEi, spironolactone– considered “main treatment classes”?

“Smoking (less than one cigarette per day; cessation of smoking less than 10 years ago was still considered smoking)”

Minor mistake, the authors probably meant “more than one cigarette” and not “less than”

2.3. Echocardiography

“A 3-5 mm pulsed-sample Doppler volume was positioned between the mitral valve tips, and trans-verse four-chamber windows at the apex were used to record the transverse flow patterns.”

PW Doppler sample volume should be 1–3 mm axial size and should be placed between mitral leaflet tips. This is according to the ASE and European association of Cardiovascular Imaging guidelines. Why was the sample volume that was used in this setting set at 3-5 mm?

“Aortic valve closure was chosen during continuous-wave Doppler recording through the aortic valve”

Chosen for what? What would be a reason to choose aortic valve closure on a CW Doppler interrogation? Please clarify.

“Segments with strain curves swinging about the zero line and segments with solely positive strain values, known as dyskinetic segments and akinetic segments, respectively, were excluded”

Should the order of dyskinetic and akinetic in the sentence be reversed?
Zero strain - akinetic
Positive strain - dyskinetic

Please double check and confirm the accuracy of the sentence.

“Two researchers who were not aware of each other's measurements and the study time point took measurements in a group of 35 randomly chosen subjects by one observer at two different times.”

Unclear statement. Please elaborate what are those 35 randomly chosen subjects, and why did they need to be chosen by an observer? i.e. Were those cases -other than the patients in this study- that were used to validate the measurements and reproducibility among the two researches who eventually did the measurements for all the study's subjects?

Also, how was the inter- and intra-observer variability investigated? What was the difference or the variability that was noted?

3.1. Patients’ Characteristics

“…, E-deceleration time, mitral regurgitant orifice area, or Patients without incidents had a mean GLS/MD at discharge of - 0.4180.2, while the others had a mean GLS/MD at dis-charge of - 0.2390.13 (p < 0.001).”

Is there a missing variable after “or” in “mitral regurgitant orifice area, or”?

Please make sure to start a new sentence starting from “Patients without incidents…” as the text is currently confusing. Also, would change the term “the others” in the sentence to a more descriptive/specific term, e.g. “those who had an incident”.

Table 1.

Therapy at hospital discharge

Why these medications were specifically looked at? Nitrates or diuretics would not be expected to change the prognosis as much as anti-platelet or statin therapy. What is the reason the latter medications were not included in the baseline characteristics?

3.2. ROC Curves to Predict Cardiac Events

“The ROC curves may be statistically compared to show that there are substantial differences between GLS/MD and E/e' (p=0.029) and GLS/MD and E/'e (p=0.015).”

GLS/MD and E/e’ is repeated twice with two different P values. Please clarify.

4. Discussion

“Our investigation found no differences in the distribution of gender, prior cardiovascular disease, cardiovascular risk factors, or medication between the group with and without cardiac events, in contrast to what is reported in the literature”

Only few medications were looked at in this study, so the sentence should not be worded as if no medications at all play a role in the outcome of these patients. Please modify.

“In our study, LVEF, LAVI, SPAP, and E/A ratio, as predictors of outcome on univariate analysis, were eliminated on multivariate analysis.”

It doesn't seem like LVEF was eliminated after multivariate analysis as the association remained statistically significant per table 2. Please clarify.

“Adding these two parameters to the regular echocardiographic exam can provide a better selection of patients for ICD therapy who do not fulfill current ICD indications”

This sentence is making a conclusion on a matter that was not directly assessed or evaluated by the study. The study did not specifically look at patients who do not fulfill ICD indications and evaluated the appropriateness of ICD therapy based on certain echocardiographic parameters. This statement should be removed from the manuscript.

“Our group had proposed the GLS/MD ratio as an important independent predictor of malignant ventricular events in patients with HF and sinus rhythm, compared to other echocardiographic parameters (LVEF, E/e’, s’, GLS and MD)”

This really can't be stated in this conclusive manner. The primary outcome in this manuscript comprised a composite of endpoints, but there is no breakdown of the composite into its individual components. What if the difference in the composite outcome is being driven by factors other than malignant arrhythmias? This needs to be elaborated and presented to provide a better understanding of the results. Otherwise, this statement should also be removed.

“The study population might not represent the entire population of patients with NSTE-ACS because the study centre served as a tertiary invasive centre.”

It is evident that the study population does not represent the entire NSTE-ACS population as there are several exclusion criteria that were mentioned in the methodology section.

Author Response

Reply to reviewer #1

We thank the reviewer for his kind remarks and excellent suggestions and comments that helped us improve this manuscript. We also uploaded the Word document where the changed are marked with red color - Please see the attachment.

-1. We agree that underlying sinus rhythm can be confusing. We didn’t include in the study group patients with atrial fibrillation because the Doppler estimation of LV filling pressures in atrial fibrillation is limited by the variability in cycle length, the absence of organized atrial activity, and the frequent occurrence of LA enlargement. Measurements from 10 cardiac cycles are most accurate, though velocities and time intervals averaged from 3 nonconsecutive beats with cycle lengths within 10% to 20% of the average heart rate and measurements from 1 cardiac cycle with an RR interval corresponding to a heart rate of 70 to 80 beats/min are still useful7.

-2. As kindly suggested by the reviewer, we tried to clarify the exclusion criteria. As suggested by Nagueh et al.7, e’ velocity is usually reduced in patients with significant annular calcification, surgical rings, mitral stenosis, and prosthetic mitral valves. It is increased in patients with moderate to severe primary MR and normal LV relaxation due to increased flow across the regurgitant valve. In these patients, Nagueh et al. recommended that the E/e’ ratio should not be used. Only patients with secondary MR were included in our study. This information was added in the Methods section (page 2, lines 45-49):

“Patients with inadequate echocardiographic images, prior myocardial infarction, open-chest surgery, cardiac pacemaker/defibrillator, mitral stenosis, severe primary MR, significant annular calcification, renal failure (serum creatinine >1.3 mg/dL), and non-cardiac illnesses with a life expectancy of less than a year were excluded from the study group.”

-3.  We completely agree that age, mean arterial pressure, and heart rate are not reported in the results text or table, which raises the question about how was this data incorporated in the analysis.

Table 1. Baseline characteristics of the study groups.

Characteristics

Event

free

(n = 201)

Cardiac events

(n = 109)

p-value

Clinical characteristics

Age, years                                                        

60.5±11.7

59.4±12.1

0.472

Mean arterial pressure, mmHg

97.9±12.8

96.5±14.5

0.261

Heart rate, beats/min

84±18

89±21

0.285

We added this data in the Table 1 (page 5, lines 4, 6, 7).

-4.  As suggested by the reviewer, we tried to clarify the main treatment classes. These data existed in the original manuscript. It seems to be a problem loading the word document. We added this data in the Table 1 (page 5, lines 30, 32, 34-36).

Therapy at hospital discharge

Beta blocker, n (%)

179 (89.0)

101 (92.6)

0.42

ACEI/angiotensin receptor antagonist, n (%)

196 (97.5)

105 (96.3)

0.81

Diuretics, n (%)

153 (76.1)

79 (72.4)

0.49

Calcium blocker, n (%)

47 (23.3)

26 (24.5)

0.85

Nitrates, n (%)                                                   

141 (70.1)

69 (63.3)

0.25

Aspirin, n (%)

201 (100)

109 (100)

1

P2Y12 inhibitor, n (%)

201 (100)

109 (100)

1

Statin,  n (%)

198 (98.5)

105 (96.3)

0.98

-5.  As suggested by the reviewer, we changed the sentence from the previous manuscript (page 3, lines 8-10):

“Smoking (less than one cigarette per day; cessation of smoking less than 10 years ago was still considered smoking)”

to

“Smoking (more than one cigarette per day; cessation of smoking less than 10 years ago was still considered smoking)”.

-6. We completely agree that PW Doppler sample volume should be 1–3 mm axial size and should be placed between mitral leaflet tips, according to the ASE and European association of Cardiovascular Imaging guidelines. In the initial project submitted to the ethics committee, a sample volume of 3-5 mm was wrongly proposed. In our study we used a 3 mm sample volume placed between mitral leaflet tips. This change was made in the new manuscript (page 3, lines 18-20):

“An axial size 3-5 mm pulsed-sample Doppler volume was positioned between the mitral valve tips, and four-chamber view was used to record the transmitral flow.”

-7. As suggested by the reviewer, we clarified aortic valve closure. We modified lines 31-33, page 3:“Aortic valve closure was chosen during continuous-wave Doppler recording through the aortic valve”  to “AVC was determined from conventional pulsed-wave Doppler signal through the aortic valve“.

-8. We modified page 3, line 36-38:  “Segments with strain curves swinging about the zero line and segments with solely positive strain values, known as akinetic segments and dyskinetic segments, respectively, were excluded”

-9. As kindly suggested by the reviewer, we tried to clarify the inter- and intra-observer variability analysis. Several changes were made:

-  page 3, line 41;

-  page 3, line 42-44;

 - page 4, lines 18;

-- page 8, line 13.

An expert echocardiographer conducted each measurement. For GLS, MD, and GLS/MD, the inter- and intra-observer variability was investigated. Two researchers who were not aware of each other's measurements and the study time point took measurements in a group of 35 randomly chosen subjects by one observer at two different times..

In 35 randomly chosen subjects of the study group, two investigators recorded measurements each, independently, a few minutes apart in the absence of the other investigator. Both observers were blinded to the each other's result.

…………

Bland-Altman analysis and intraclass correlation coefficients were used to quantify the intra-observer variability and inter-observer variability for GLS, MD, and GLS/MD. Utilizing SPSS Statistics for Windows, data was analyzed (IBM SPSS Statistics for Windows, Version 21.0. Armonk, NY: IBM.).

…………

3.6. Reproducibility

Bland-Altman analysis showed that The intra- and inter-observer agreements were good for GLS/MD, GLS and MD measurements in 35 patients with ischemic heart disease. For inter-observer and intra-observer variability, the intraclass correlation coefficients were 0.92 and 0.89 for GLS/MD, 0.94 and 0.93 for GLS, and 0.92 and 0.90 for MD, respectively.

-10. We modified page 4, lines 42-44: “…, E-deceleration time, mitral regurgitant orifice area, or mitral regurgitant volume. Patients without incidents had a mean GLS/MD at discharge of - 0.418±0.2, while those who had an incident had a mean GLS/MD at dis-charge of - 0.239±0.13 (p < 0.001).”

-11.  As suggested by the reviewer ♯1, we corrected in Table 1 the Therapy at hospital discharge. We added the missing data in the Table 1 (page 5, lines 30, 32, 34-36).

Therapy at hospital discharge

Beta blocker, n (%)

179 (89.0)

101 (92.6)

0.42

ACEI/angiotensin receptor antagonist, n (%)

196 (97.5)

105 (96.3)

0.81

Diuretics, n (%)

153 (76.1)

79 (72.4)

0.49

Calcium blocker, n (%)

47 (23.3)

26 (24.5)

0.85

Nitrates, n (%)                                                   

141 (70.1)

69 (63.3)

0.25

Aspirin, n (%)

201 (100)

109 (100)

1

P2Y12 inhibitor, n (%)

201 (100)

109 (100)

1

Statin,  n (%)

198 (98.5)

105 (96.3)

0.98

-12.  As kindly suggested by the reviewer, we corrected the comparison of the ROC curves. We changed page 6 lines 20-22 from “The ROC curves may be statistically compared to show that there are substantial differences between GLS/MD and E/e' (p=0.029) and GLS/MD and E/'e (p=0.015).” to “The ROC curves may be statistically compared to show that there are substantial differences between GLS/MD and E/e' (p=0.029) and GLS/MD and MD (p=0.015).”

-13.  In the Discussion section we changed lines 18-20, page 8 of the revised version of the manuscript: “Our investigation found no differences in the distribution of gender, prior cardiovascular disease, or cardiovascular risk factors, or medication between the group with and without cardiac events, in contrast to what is reported in the literature”  

-14. We corrected the statement (page 8, line 28 ): “In our study, LVEF, LAVI, SPAP, and E/A ratio, as predictors of outcome on univariate analysis, were eliminated on multivariate analysis.”  LVEF was excluded.

-15. We completely agree that the sentence “Adding these two parameters to the regular echocardiographic exam can provide a better selection of patients for ICD therapy who do not fulfill current ICD indications” is making a conclusion on a matter that was not directly assessed or evaluated by the study. This statement has been removed from the corrected manuscript.

-16. We reanalyzed the statement “Our group had proposed the GLS/MD ratio as an important independent predictor of malignant ventricular events in patients with HF and sinus rhythm, compared to other echocardiographic parameters (LVEF, E/e’, s’, GLS and MD)” (page 10, line 9-12 in the previous version of the manuscript). This really can't be stated in this conclusive manner. As suggested by the reviewer, we removed this statement.

-17. We completely agree that it is evident that the study population does not represent the entire NSTE-ACS population as there are several exclusion criteria that were mentioned in the methodology section. The statement (page 10, lines 19-20)  “The study population might not represent the entire population of patients with NSTE-ACS because the study centre served as a tertiary invasive centre.”was removed.

Reviewer 2 Report

This paper evaluates the prognostic significance of GLS and MD in NSTE-ACS patients. The Authors introduce the predictive utility of the novel GLS/MD 2-dimensional strain index in this population. The Authors conclude that this index represents a strong indicator of clinical fate.

Methodology is outstanding

The paper is well written

Author Response

We thank the reviewer for his kind comments that give us the strength to continue research in this direction. Larger investigations are therefore required to confirm our findings.

Reviewer 3 Report

The author and team investigate a very important and well thought of question in the management and risk stratification of patients with acute coronary syndrome. It has strong clinical and scientific relevance and the process of investigation was very thorough. This study most certainly merits publication without any significant changes. i see no glaring flaws and feel confident in recommending publication of this study without any major changes whatsoever. As mentioned in the study, small sample size is biggest limitation of the study. However, the study is hypothesis generating and would warrant future prospective studies. 

Author Response

We thank the reviewer for his kind remarks and comments that helped us improve this manuscript. We careful revised the orthography and grammar of our manuscript.